# Research on Control of Intelligent Vehicle Human-Simulated Steering System Based on HSIC

**Haobin Jiang** [1,*]**, Huan Tian** [1] **, Yiding Hua** [2,3] **and Bin Tang** [4]

[1] School of Automotive and Traffic Engineering, Jiangsu University, Zhenjiang 212013, China; tianhuan16@163.com

[2] China Automotive Research (Tianjin) Automotive Engineering Research Institute Co., Ltd., Tianjin 300300, China; dingyihua0209@163.com

[3] School of Mechanical Engineering, Tianjin University, Tianjin 300072, China

[4] Automotive Engineering Research Institute, Jiangsu University, Zhenjiang 212013, China; tangbin@ujs.edu.cn

\* Correspondence: jianghb@ujs.edu.cn; Tel.: +86-0511-8878-0272

**Abstract:** The experienced drivers with good driving skills are used as objects of learning, and road steering test data of skilled drivers are collected in this article. First, a nonlinear fitting was made to the driving trajectory of skilled driver in order to achieve human-simulated control. The segmental polynomial expression was solved for two typical steering conditions of normal right-steering and U-turn, and the *hp* adaptive pseudo-spectral method was used to solve the connection problem of the vehicle segmental driving trajectory. Second, a new Electric Power Steering (EPS) system was proposed, and the intelligent vehicle human-simulated steering system control model based on human simulated intelligent control (HSIC) was established in Simulink/Carsim joint simulation environment to simulate and analyze. Finally, in order to further verify the effectiveness of the proposed algorithm in this article, an intelligent vehicle steering system test bench with a steering resistance torque simulation device was built, and the dSPACE rapid prototype controller was used to realize human-simulated intelligent control law. The results show that the human-simulated steering control algorithm is superior to the traditional proportion integration differentiation (PID) control in the tracking effect of the steering characteristic parameters and passenger comfort. The steering wheel angle and torque can better track the angle and torque variation curve of real vehicle steering experiment of the skilled driver, and the effectiveness of the intelligent vehicle human-simulated steering control algorithm based on HSIC proposed in this article is verified.

**Keywords:** Intelligent vehicle; skilled driver; *hp* adaptive pseudo-spectral method; HSIC; EPS

## 1. Introduction

The intellectualization of automobiles is reflected in replacing manual operation with automatic driving. The controllable and predictable behavior and driving state of automobiles can not only make up for the deficiency of human sensory ability, but also reduce the disutility of driving and eliminate traffic accidents caused by human factors. Consequently, "zero casualty and zero congestion" of traffic can be realized based on the travel path planned according to the real-time traffic information. Therefore, the next generation of intelligent automobiles is safe, efficient, and energy efficient [1,2]. It is of great significance to study intelligent vehicles, which have become the focus of attention in the global automotive industry.

The automatic driving control technology of intelligent vehicle faces many difficulties in substituting human drivers completely within all conditions range and being accepted by

consumers [3,4]. For an intelligent vehicle, when it only controls the steering wheel angle according to the signal of the Electric Power Steering (EPS) angle sensor, it does not control the EPS. When the output torque of the motor is invariant, the steering wheel angle or the rotation speed fluctuates, resulting in steering wheel jitter. If the damping of the EPS motor (reverse torque) is not controlled during the steering process of the large angle working condition, the steering wheel will be returned rapidly, causing a large yaw rate of the vehicle body. As a result, the steering quality of the intelligent vehicle will be deteriorated. There is even instability at high speed, which jeopardizes the safety of driving. Therefore, it is necessary to simulate the driving behavior of the real driver. Meanwhile, an intelligent control algorithm with a higher degree of intelligence is applied to the intelligent driving vehicle, it will lead the driving level of the intelligent vehicle is as close as possible to the handling quality of the human driver.

In the research process of intelligent vehicle steering control, the classical control theory has been widely used in the 1980s. Broggi used the P controller in the design process of the intelligent vehicle control system [5]. Recent studies have shown that the classic PID control algorithm has certain limitations. For example, the parameters need to be artificially set, and cannot be adaptively adjusted according to different driving conditions. Therefore, the control accuracy can hardly be guaranteed [6]. According to the anti-step sliding mode theory, Hu Ping et al. designed a control algorithm for the coordinated vertical and horizontal motions of intelligent vehicles and verified the consistent and bounded convergence of tracking errors through the Lyapunov function [7]. Based on the strong coupling, nonlinearity, and parameter uncertainty of intelligent vehicles, Jinghua and Yugong proposed a collaborative control strategy with hierarchical structure [8]. Xijun and Huiyan established a lateral path tracking control system for intelligent vehicles. The control system mainly designed the corresponding feedback control logic based on the heading deviation [9]. According to the analysis, the above references cannot simulate the driving behavior of real drivers. Naranjo et al., established an overtaking system for an automated vehicle equipped with path tracking and lane change functions, so that human behavior and response during overtaking can be stimulated through fuzzy controllers [10]. However, the above method only controls the steering wheel angle and does not consider the torque of the steering wheel, it leads fluctuate inevitably for the steering wheel angle or the rotation speed, thereby, steering wheel occurs shake.

Human simulated intelligent control (HSIC) is a logic-based intelligent control algorithm. The main goal of HSIC is not the controlled object, but how the controller itself simulates the behavior of the expert. HSIC has been successfully applied to the MR semi-active suspension system [11,12] and the combustion process of gas heating furnace [13]. Intelligent vehicle automatic steering system was a nonlinear time-varying complex system. There were many uncertain factors, and the difficulty of control strategy design increased. In view of the fact that the HSIC algorithm can better simulate the excellent control characteristics unique to human control behavior, and the HSIC has been successfully applied to some difficult objects in the industrial field, this provides a relatively novel idea for the design of intelligent vehicle automatic steering system controller.

Therefore, subjects who were skilled in driving were selected as the objects of learning and simulation. Moreover, the driving trajectory data of skilled drivers under different vehicle models, vehicle speeds, and steering conditions were collected. In order to do human-simulated control, the driving trajectory of skilled drivers was simulated. The segmental polynomial expression of the above-mentioned driving trajectory was solved under the two typical steering conditions of normal right-steering and U-turn, and the *hp* adaptive pseudo-spectral method was used to solve the connection problem of the vehicle segmental driving trajectory. Meanwhile, a novel EPS steering system was proposed. The steering system dynamics model and motor model were established. In the Simulink/Carsim joint simulation environment, the intelligent vehicle human-simulated steering system control model based on HSIC was established to carry out the simulation analysis. In order to further verify the validity and reliability of the human-simulated intelligent control law, an intelligent vehicle steering system test bench with a steering resistance torque simulation device was built, and the

dSPACE rapid prototyping platform was used as the actual control of the intelligent vehicle steering system platform. The device constitutes a control system bench test platform to comprehensively analyze the system control performance.

The structure of the paper is as follows. Section 2 presents polynomial fitting of vehicle driving trajectory of skilled driver, and the *hp* adaptive pseudo-spectral method is used to solve the connection problem of the vehicle segmental driving trajectory. Design and performance simulation of intelligent vehicle steering controller based on human-simulated intelligence theory is shown in Section 3. In Section 4, the rapid control prototype test of human-simulated steering system will be highlighted. Finally, the conclusions are given in Section 5.

## 2. Polynomial Fitting of Vehicle Driving Trajectory

### 2.1. Data Collection

In this experiment, five driving school coaches of different years of experience and genders were selected as the test drivers, as shown in Table 1. In the real vehicle test, left/right steering condition and U-turn condition were proposed. Each driver conducted three tests under each working condition. The average value of the three tests was used as reference data. The speed of the vehicle was constant in the test.

**Table 1.** Driver information.

|  | Ages (Years) | Driving Experience (Years) | Gender |
|---|---|---|---|
| Driver 1 | 55 | 33 | Female |
| Driver 2 | 28 | 10 | Male |
| Driver 3 | 53 | 31 | Male |
| Driver 4 | 46 | 22 | Male |
| Driver 5 | 53 | 21 | Male |

In the process of collecting skilled drivers' steering tests, the sensors used mainly include: S-Motion dual-axis optical speed sensor, Ki MSW powerful steering wheel sensor (range 250 Nm), distribution box, and SDI-600GI GPS/INS. The physical quantity and accuracy measured by the sensor are shown in Table 2.

**Table 2.** Sensor name and accuracy.

| Sensor Name | The Measured Physical Quantity | Unit | Accuracy |
|---|---|---|---|
| S-Motion biaxial optical speed sensor | Velocity | km/h | $<\pm0.3$ |
|  | Roll angle | deg | $<\pm0.01$ |
|  | Lateral acceleration | $m/s^2$ | $<\pm0.15$ |
|  | Yaw rate | deg/s | $<\pm0.3$ |
| Ki MSW steering wheel force sensor | Steering angle | deg | $<0.05$ |
|  | Steering torque | Nm | $<0.1$ |
|  | Steering angular velocity | deg/s | $<\pm0.3$ |
| SDI-600GI model GPS/INS | Longitude, latitude and elevation | cm (longitude $1'' \approx 30.7$ m) | $<10$ |

### 2.2. The Trajectory Fitting of Normal Left/right Steering Condition

The driving trajectory of vehicles in the normal left/right steering condition is shown in Figure 1a. In order to accurately represent the steering characteristics of the excellent driver, the segmental polynomial was used to fit the trajectory of vehicles under normal left/right steering condition. The first and second segment fitting curves are shown in Figure 1b,c. $R^2$ is 0.9942 and 0.993, respectively, which is close to 1, indicating the high fitting accuracy.

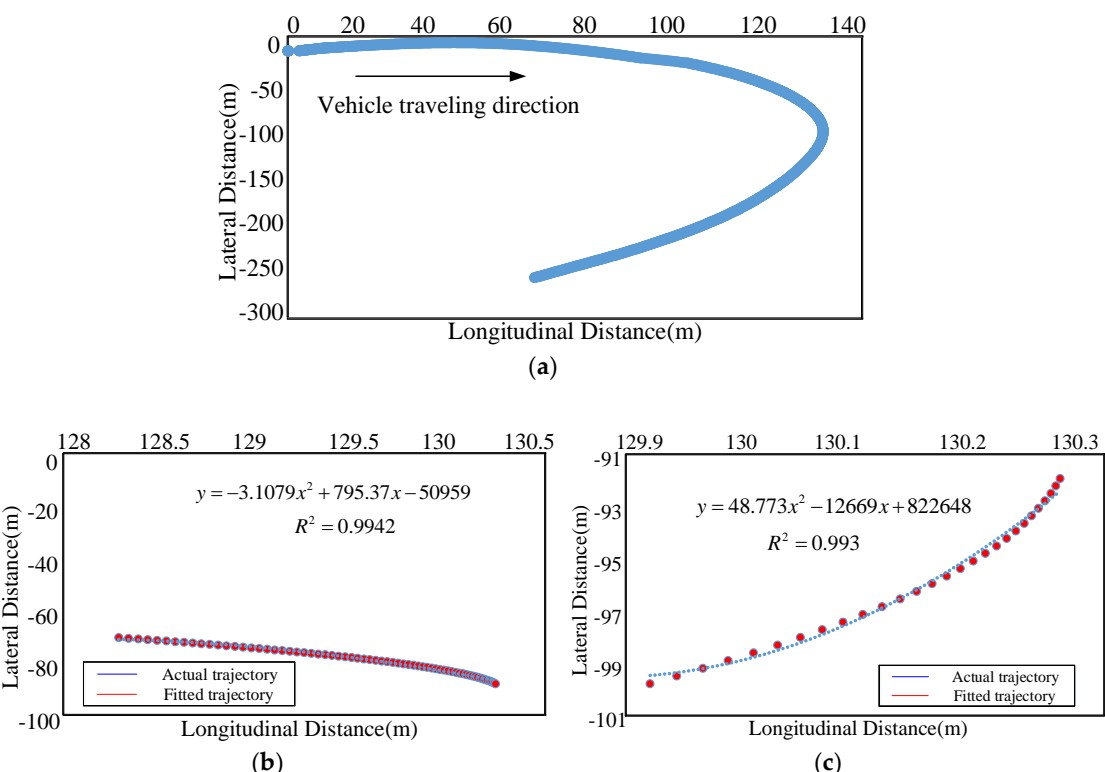

**Figure 1.** Trajectory fitting under normal right-steering: (**a**) Full trajectory; (**b**) The first paragraph trajectory; (**c**) The second paragraph trajectory.

### 2.3. The Trajectory Fitting of U-Turn Condition

The vehicle trajectory under the U-turn condition is shown in Figure 2a. In order to accurately represent the steering characteristics of the excellent driver, the segmental polynomial was used to fit the vehicle trajectory under U-turn condition. The first and second stage fitting curves are shown in Figure 2b,c. $R^2$ is 0.9844 and 0.9927, respectively, which is close to 1, indicating the high fitting accuracy.

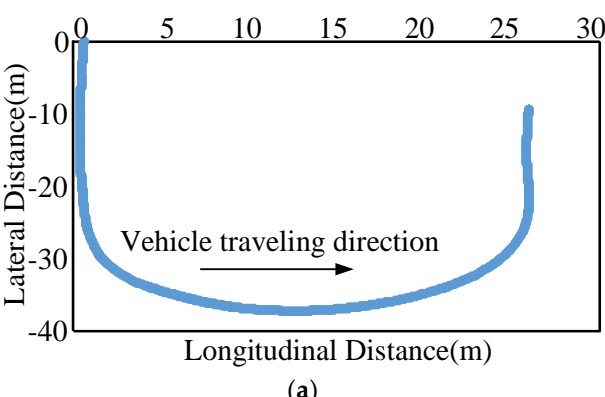

**Figure 2.** *Cont.*

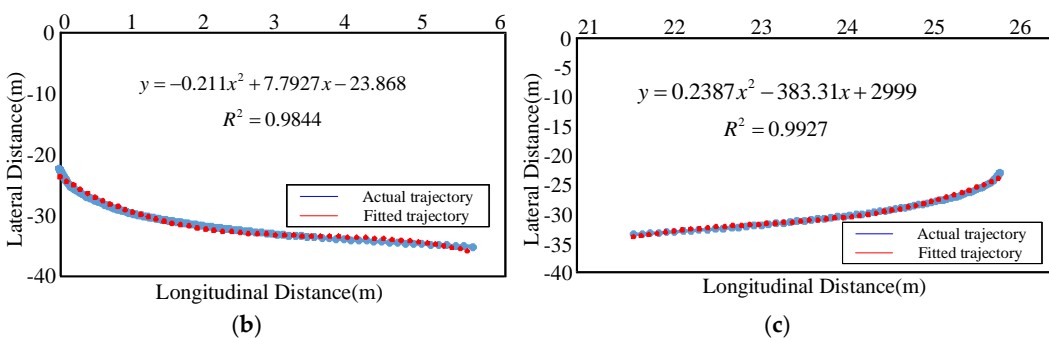

**Figure 2.** Trajectory fitting under U-turn: (**a**) Full trajectory; (**b**) The first paragraph trajectory; (**c**) The second paragraph trajectory.

### 2.4. Segmental Function Connection Point Processing

In order to accurately represent the steering characteristics of the excellent driver, the normal left/right steering condition and U-turn condition can be fitted using a segmental function, the fragments of trajectory is changed by the middle value of the trajectory. The highest order of the polynomial is 2 times. Therefore, the path function expression is expressed as follows:

$$\begin{cases} y_1 = a_2x^2 + a_1x + a_0 \\ y_2 = b_2x^2 + b_1x + b_0 \\ y_3 = c_2x^2 + c_1x + c_0 \end{cases} \tag{1}$$

where $a_2 \sim a_0$, $b_2 \sim b_0$ and $c_2 \sim c_0$ are the various coefficients that need to be determined. Once the various coefficients are determined, the vehicle driving trajectory under various conditions can also be determined.

The convergence problem of segmented trajectory directly affects the accuracy of vehicle trajectory fitting. The advantage of pseudo-spectral method [14] is that the global orthogonal polynomial is used to approximate the state quantity and control quantity at the joint. The corresponding constraint at the switching point can be introduced to handle the convergence problem of segmented trajectories. A trajectory is assumed to consist of two paths. The state quantity and the control quantity at the joint point are expressed as follows:

$$\xi(t_e^{(1)}) - \xi(t_0^{(2)}) = 0, u(t_e^{(1)}) - u(t_e^{(2)}) = 0 \tag{2}$$

Which is:

$$\varphi\left(t_0^{(2)}\right) = \varphi\left(t_e^{(1)}\right), v\left(t_0^{(2)}\right) = v\left(t_e^{(1)}\right), \delta\left(t_0^{(2)}\right) = \delta\left(t_e^{(1)}\right) \tag{3}$$

where, $\xi$ is the state quantity, $u$ is the control quantity, $\varphi$ is the yaw angle of the vehicle, $v$ is vehicle speed, $\delta$ is the steering wheel angle, $t_0^{(1)}$ is start time of the first phase path, $t_e^{(1)}$ is termination time of the first phase path, $t_0^{(2)}$ is start time of the second phase path, and $t_e^{(2)}$ is termination time of the second phase path.

Since the pseudo-spectral method needs to convert the time zone to $[-1, 1]$, it is necessary to transform the time domain of the vehicle travel trajectory. The time zone is divided into multiple sub-intervals, and each sub-interval is converted into nonlinear programming problem (NLP) solution. Therefore, $K$-1 nodes are selected, and the optimal control problem is divided into $K$ sub-intervals at $t \in \left[t_0^{(2)}, t_e^{(2)}\right]$, namely, $t_0^{(2)} < t_1 < \cdots < t_k = t_e^{(2)}$. Since choosing the location of the connection point is an iterative process, the selection of $t_0^{(2)}$ value is also an iterative process.

$$\hat{t}_0^{(2)} = t_0^{(2)} + \gamma \tag{4}$$

where $\gamma$ is the given time frequency.

For any subinterval $k$, the time zone is transformed from $t \in [t_{k-1}, t_k]$ to $\tau \in [-1, 1]$ through the following formula.

$$\tau = \frac{2t - (t_{k-1} + t_k)}{t_k - t_{k-1}}, (t_{k-1} < t < t_k) \tag{5}$$

The *hp* adaptive pseudo-spectral method checks the number and distribution of discrete points after each optimization calculation. When the calculation accuracy of a discrete interval does not meet the requirements, the match point number *h* in the interval and the dimension *p* of the global interpolation polynomial were adaptively adjusted by the *hp* adaptive method. Then the next optimization calculation begins until the residuals meet the requirements. The algorithm flow is shown in Figure 3.

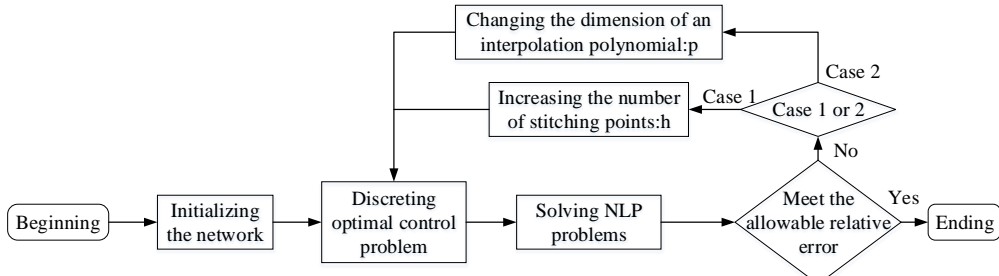

**Figure 3.** Adaptive pseudo-spectral calculation process.

## 3. Design and Performance Simulation of Intelligent Vehicle Steering Controller Based on Human-Simulated Intelligence Theory

### 3.1. The Modelling of Intelligent Vehicle Steering System

This paper proposes a novel EPS system, which is a man-machine-assisted electric power steering system. The structural principle is shown in Figure 4.

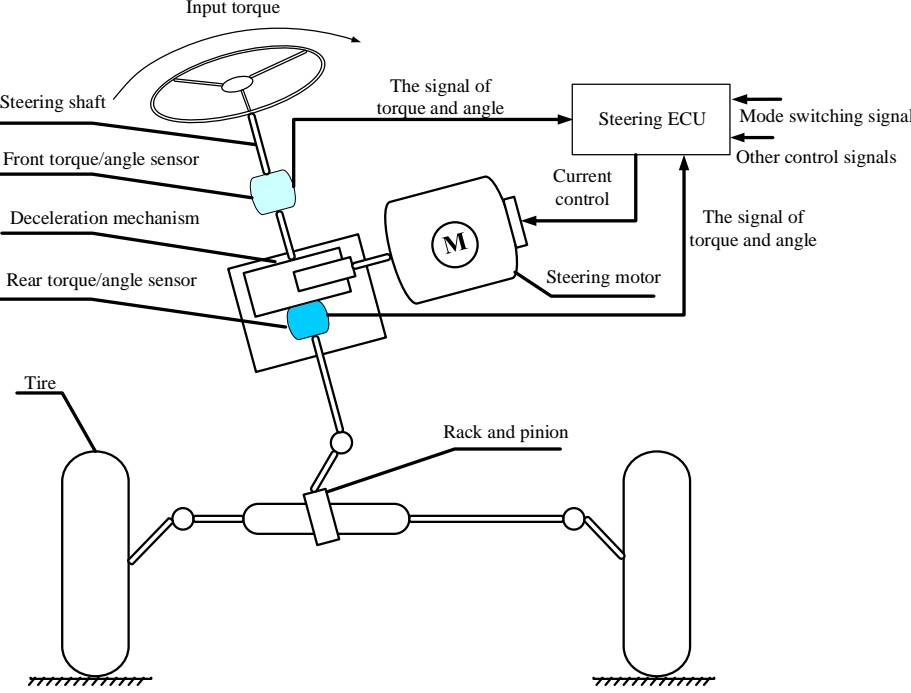

**Figure 4.** Novel Electric Power Steering (EPS) system composition.

The EPS system is a complex nonlinear system. In order to facilitate to analysis its dynamics problem, this paper established an ideal EPS system model [15–17].

The force analysis of the novel EPS:

$$J_s\ddot{\theta}_s + B_s\dot{\theta}_s = GT_m - T_{sen} \tag{6}$$

$$m_r\ddot{x}_r + b_r\dot{x}_r = \frac{T_w}{r_p} - F_{TR} \tag{7}$$

where $J_s$ is the moment of inertia of the steering wheel and steering shaft, $B_s$ is the viscous damping coefficient of the steering shaft, $\theta_s$ is the rotation angle of the steering shaft, $T_m$ is the motor torque, $T_{sen}$ is the reaction torque of the torsion bar, $G$ is the speed reduction ratio of the reduction mechanism, $m_r$ is the equivalent mass of the rack and pinion, $b_r$ is the damping coefficient of the rack, $x_r$ is the displacement of the rack, $T_w$ is the reaction torque acting on the output shaft, $r_p$ is the pinion radius, and $F_{TR}$ is the axial force acting on the rack.

The motor model is:

$$J_m\ddot{\theta}_m + B_m\dot{\theta}_m = T_m - T_L \tag{8}$$

$$V_m = R_mI_m + L_m\dot{I}_m + K_e\dot{\theta}_m \tag{9}$$

where $J_m$ is the moment of inertia of the motor and clutch, $B_m$ is the viscous damping coefficient of the motor, $\theta_m$ is the rotation angle of the motor, $T_L$ is the load torque of the motor, $R_m$ is the motor armature resistance, $I_m$ is the motor armature current, $L_m$ is the armature winding inductance, $K_e$ is the counter-electromotive force constant, and $V_m$ is the motor terminal voltage.

### 3.2. The Design of Intelligent Vehicle Human-Simulated Steering Controller

Intelligent vehicle automatic steering system is a nonlinear time-varying complex system. There are many uncertain factors, which enhances the difficulty of control strategy design. To analyze the operating characteristics of the steering system, the steering process is divided into three stages, namely, the steering start stage, the steering maintenance stage, and the return stage. Each process includes two aspects of angle and torque, namely, the angle and torque change phase, the angle and torque maintenance phase, and the angle and torque return stage. The changes of each stage includes the following modes, such as the position expression of target trajectory in the deviation phase plane $e - \dot{e}$, as shown in Figure 5.

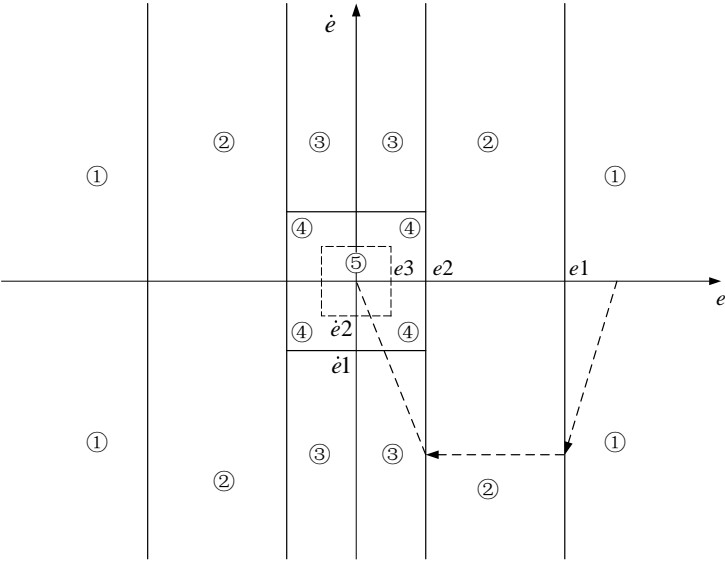

**Figure 5.** Steering system (angle and torque) phase plane design.

When the deviation is large, corresponding to area ①, the largest possible control effect, namely, pound-pound control is adopted.

When the deviation and the change rate of the deviation are small (satisfying the requirements), corresponding to area ⑤, in order to eliminate the error, PID control is adopted.

If the deviation is large, corresponding to area ② proportional modal control is adopted. In order to ensure the low deviation change speed, weak differential control can be introduced on the basis of proportional mode.

In the process of reducing the deviation, if the deviation change speed is lower than or equal to the predetermined speed, corresponding to area ④, the proportional mode plus a differential mode control is adopted. In the process of reducing the deviation, if the deviation change speed is greater than the predetermined speed, corresponding to ③, strong differential control is introduced to the proportional mode, so that the deviation change speed is reduced as fast as possible.

Therefore, according to the deviation and deviation change speed, the operation control level can be divided into five modes:

Control mode 1: $u_n = \text{sgn}(e_n) \cdot U_{\max}, |e_n| > e_1$
Control mode 2: $u_n = k_{p2} \cdot e + k_{d2} \cdot \dot{e}, |e_n| < e_1 \cap |e_n| > e_2$
Control mode 3: $u_n = k_{p3} \cdot e + k_{d3} \cdot \dot{e}, |e_n| < e_2 \cap |\dot{e}_n| \geq |\dot{e}_1|$
Control mode 4: $u_n = k_{p4} \cdot e + k_{d4} \cdot \dot{e}, |e_n| < e_2 \cap |\dot{e}_n| < |\dot{e}_1| \cap \neg(|e_n| < e_3 \cap |\dot{e}_e| < |\dot{e}_2|)$
Control mode 5: $u_n = k_{p5} \cdot e + k_{d5} \cdot \dot{e} + k_{i5} \int edt, |e_n| < e_3 \cap |\dot{e}_n| < |\dot{e}_2|$

where $e$ is the system deviation, $\dot{e}$ is the rate of deviation change, $u_n$ is the output of the controller, $U_{\max}$ is the maximum control output, and $e_n$ is the $n$-th peak of the deviation.

According to the above analysis, the human-simulated control system of the steering system is established in the MTLAB/Simulink environment, which mainly includes five control modes, deviation differential links, time-delay links, and controlled object design, as shown in Figure 6.

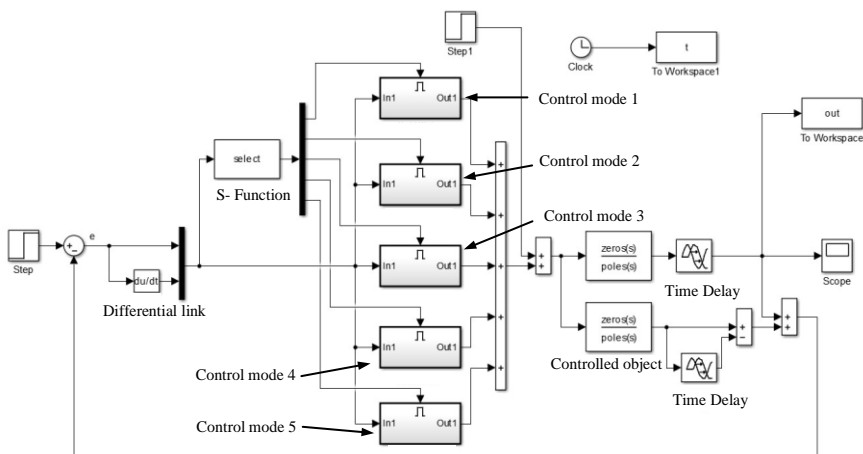

**Figure 6.** Steering system human-simulated control system based on MTLAB/Simulink.

The overall system control model based on Simulink/Carsim joint simulation is established, as shown in Figure 7.

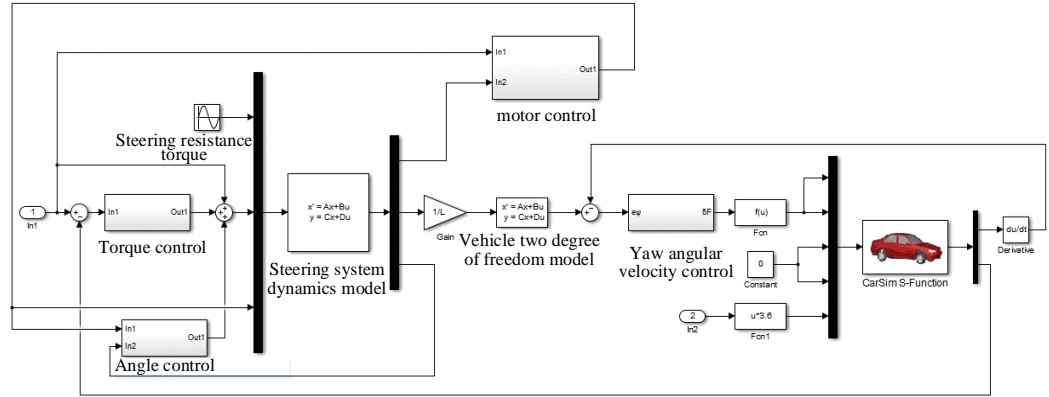

**Figure 7.** Overall control system based on Simulink/Carsim.

### 3.3. The Performance Simulation Analysis of Human-Simulated Steering Control

The vehicle model parameters are shown in Table 3.

**Table 3.** Vehicle model parameters.

| Parameter Name | Numerical Value | Unit |
|---|---|---|
| Vehicle mass | 1558 | kg |
| Length/width/high | 4915/1845/1470 | mm |
| Wheelbase | 2775 | mm |
| Front wheel distance/rear wheel distance | 1585/1580 | mm |
| The distance from the vehicle centroid to the front and rear axles | 1110/1665 | mm |
| Moment of inertia of the vehicle around the X, Y, Z-axis | 671.3/1972.8/2315.3 | kg·m$^2$ |
| Tire specifications | 225/50 R17 | |

For the accurate acquisition of human-simulated intelligent control parameters, there is no uniform and effective method. In this paper, the tested and tried method is used to obtain the control parameters. By comparing the system control performance under different sequence control parameters, a group with relatively good control performance is selected. The parameters are used as the actual control parameters of the system. The relevant parameters of the controller are set as follows: $U_{\max} = 5$, $k_{p2} = 5$, $k_{d2} = 2$, $k_{p3} = 5$, $k_{d3} = 3$, $k_{p4} = 10$, $k_{d4} = 0.8$, $k_{p5} = 50$, $k_{d5} = 9$.

According to Figure 8, under normal right-steering condition, the steering wheel angle and torque obtained by human-simulated intelligent control, whether at a lower vehicle speed (20 km/h) or a higher vehicle speed (50 km/h), can better follow the steering wheel angle and torque of the reference trajectory and have good control effect. The traditional PID control algorithm can effectively follow the reference curve in the initial stage. However, with the increase of the steering wheel angle, obvious control lag and control overshoot occur in PID control. Therefore, the human-simulated intelligent control algorithm has much better control effect than the traditional PID control algorithm. In addition, the yaw rate of the human-simulated intelligent control is significantly lower than that of the traditional PID, indicating that the human-simulated intelligent control can effectively improve the comfort of passengers. It can be found from the low lateral deviation that the vehicle is always in a stable state. Therefore, the proposed control method can effectively improve the comfort of the passengers.

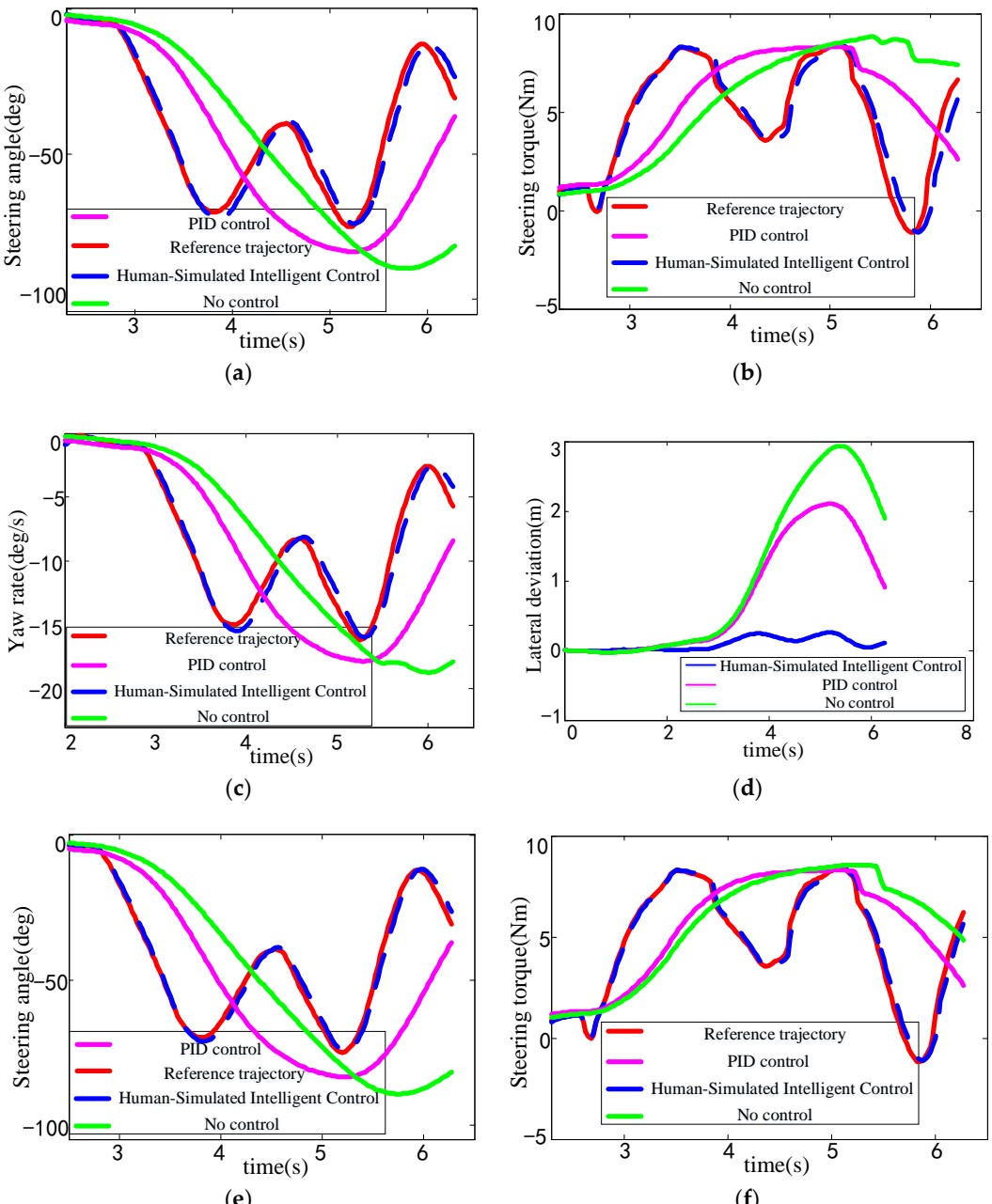

**Figure 8.** Comparison and analysis of control results under normal right-steering condition: (**a**) 20 km/h, comparison of steering wheel angle; (**b**) 20 km/h, comparison of steering wheel torque; (**c**) 20 km/h, comparison of yaw rate; (**d**) 20 km/h, comparison of lateral deviation; (**e**) 50 km/h, comparison of steering wheel angle; (**f**) 50 km/h, comparison of steering wheel torque.

According to Figure 9, whether at the lower speed (20 km/h) or the higher speed (30 km/h), the steering wheel angle and torque obtained by the human-simulated intelligent control can better follow steering wheel angle and torque under the U-turn condition. The traditional PID control algorithm can effectively follow the reference curve in the initial stage. However, with the increase of the steering wheel angle, the PID control shows slightly control lag and control overshoot. This is because during the normal right-steering condition, the turning radius is relatively large, and the turning is only once. However, in the process of U-turn condition, the vehicle speed is slower, the turning radius is smaller, and the turning occurs two times. When the vehicle speed is low, the PID control effect will be better, so the PID lag in the U-turn condition compared

with the normal right-steering condition is not obvious. Therefore, the control effect based on human-simulated intelligent control calculation is better than the traditional PID control algorithm at imitating experienced drivers. It can be seen from the lateral deviation that the vehicle-based intelligent control always maintains a low lateral deviation, indicating that the vehicle is always in a stable state. Therefore, the proposed control method can effectively improve the comfort of passengers.

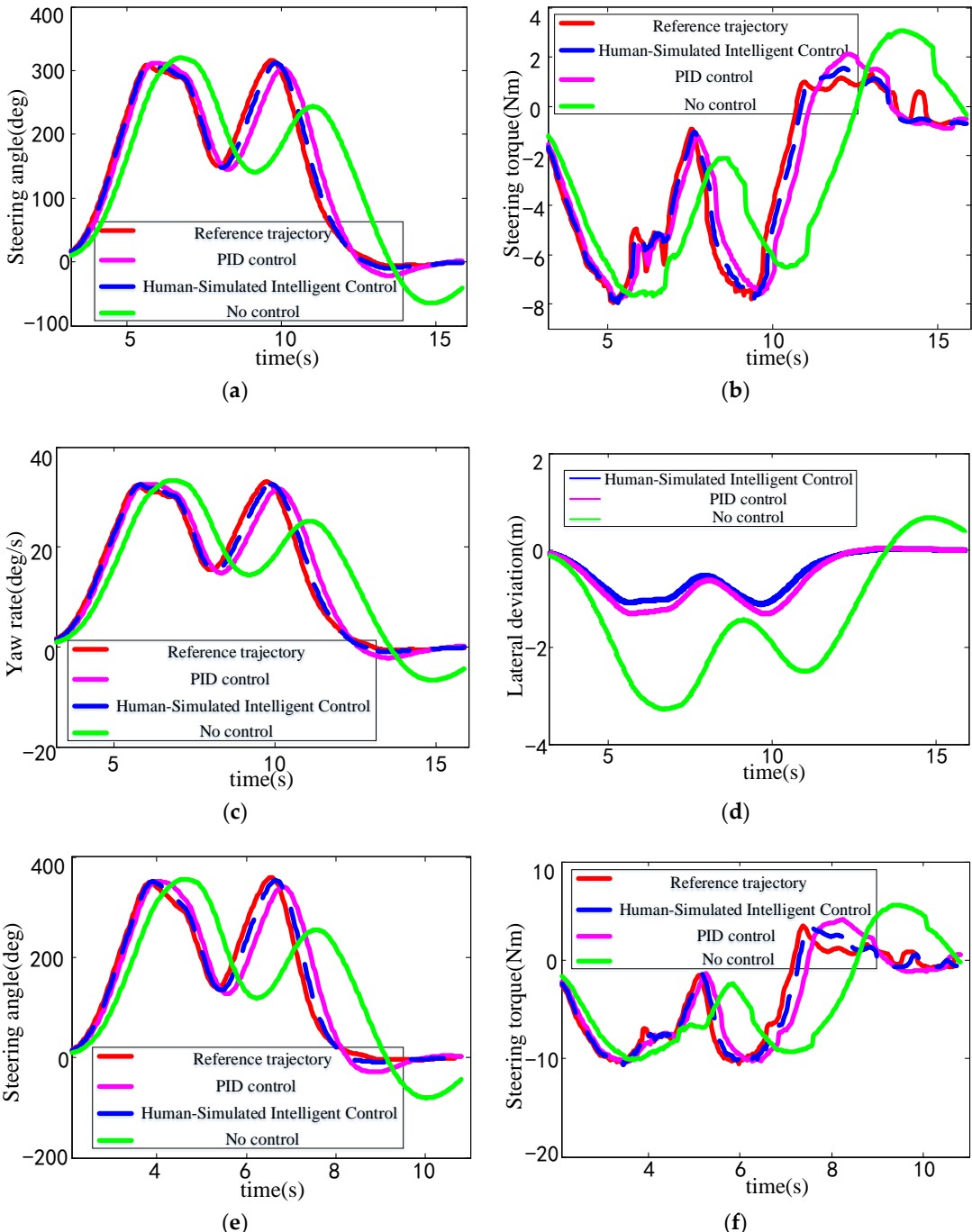

**Figure 9.** Comparison and analysis of control results under U-turn condition: (**a**) 20 km/h, comparison of steering wheel angle; (**b**) 20 km/h, comparison of steering wheel torque; (**c**) 20 km/h, comparison of yaw rate; (**d**) 20 km/h, comparison of lateral deviation; (**e**) 30 km/h, comparison of steering wheel angle; (**f**) 30 km/h, comparison of steering wheel torque.

## 4. The Rapid Control Prototype Test of Human-Simulated Steering System

### 4.1. The Implementation of Control Scheme

In order to verify the test performance of the human-simulated steering system based on the dSPACE rapid prototyping platform, the control scheme was designed. The control implementation employed herein is illustrated in Figure 10. First of all, the human-simulated intelligent control algorithm was compiled to automatically generate the control code. Secondly, the above control code was downloaded into the dSPACE hardware. Finally, the angle/torque sensor signal transmission and the actuator were used to drive the controlled object and realize the connection between human-simulated steering system and the dSPACE rapid prototype controller. In this way, a test platform for the intelligent vehicle human-simulated steering control system was constructed. The bench test was verified based on the test platform.

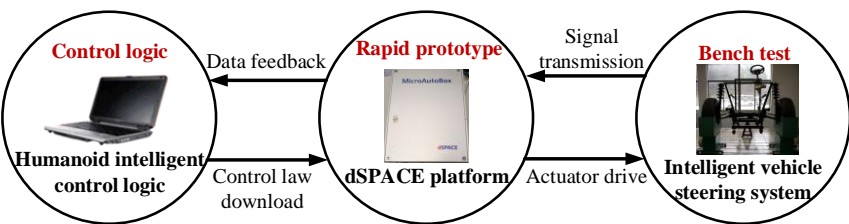

**Figure 10.** Implementation plans of the control system.

The simulation environment of the vehicle model based on Simulink/Carsim was used to more accurately test the human-simulated steering control system under two typical conditions. Combined with the dSPACE rapid prototype control platform, a hardware human steering system test bench was built on the basis of the EPS of a passenger car and the steering column. The overall structure of the test bench is shown in Figure 11.

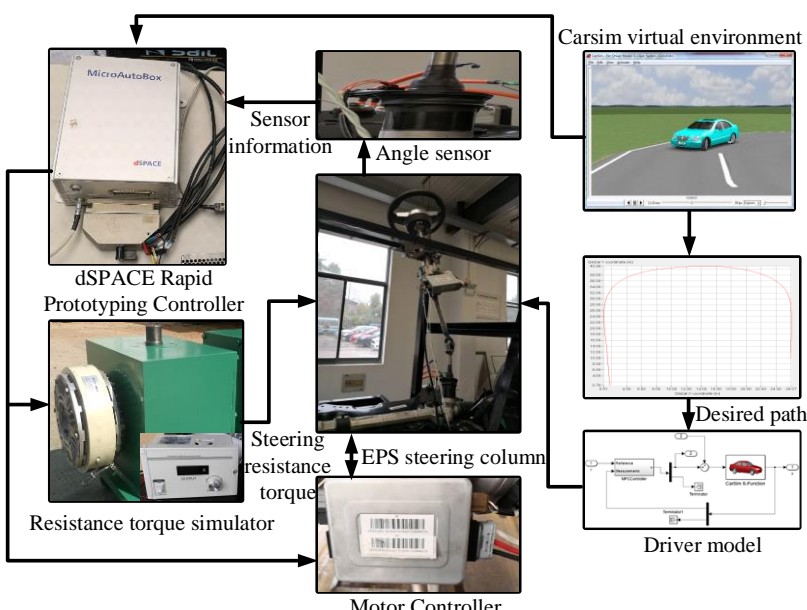

**Figure 11.** Test bench overall structure.

The human-simulated steering system rapid control prototype test bench is shown in Figure 12.

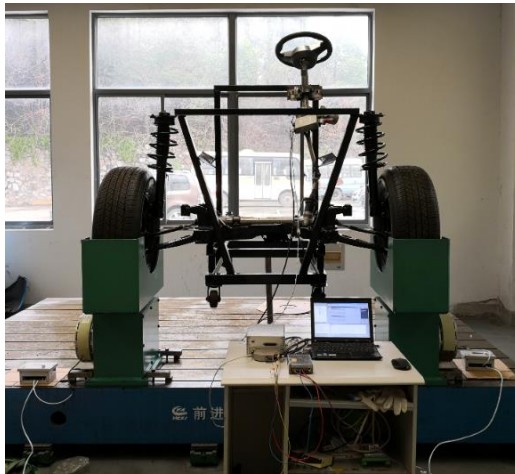

**Figure 12.** Human-simulated Steering System Rapid Control Prototype Test Bench.

*4.2. Test Results and Performance Analysis*

In order to further verify the actual performance of the control system, the collected real vehicle test data of the skilled driver and the results of the bench test were compared and analyzed. The normal right-steering and U-turn test conditions are given respectively, as shown in Figures 13 and 14.

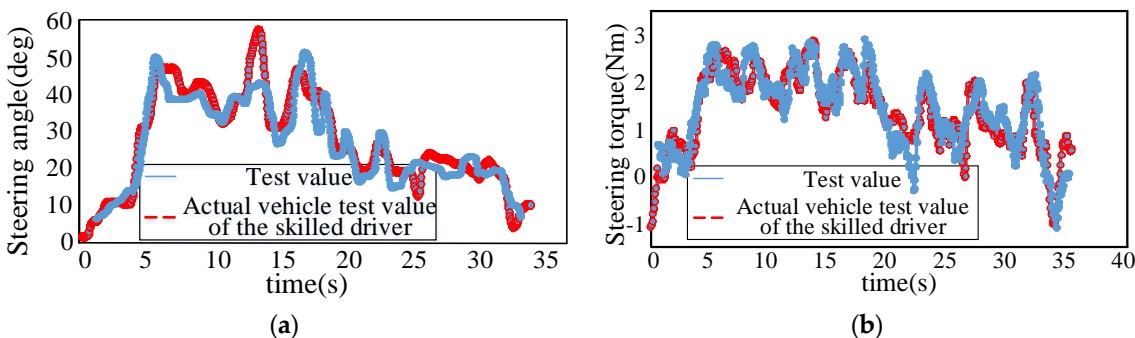

**Figure 13.** Comparison of test results under normal right-steering condition. (**a**) Comparison of steering wheel angle; (**b**) Comparison of steering wheel torque.

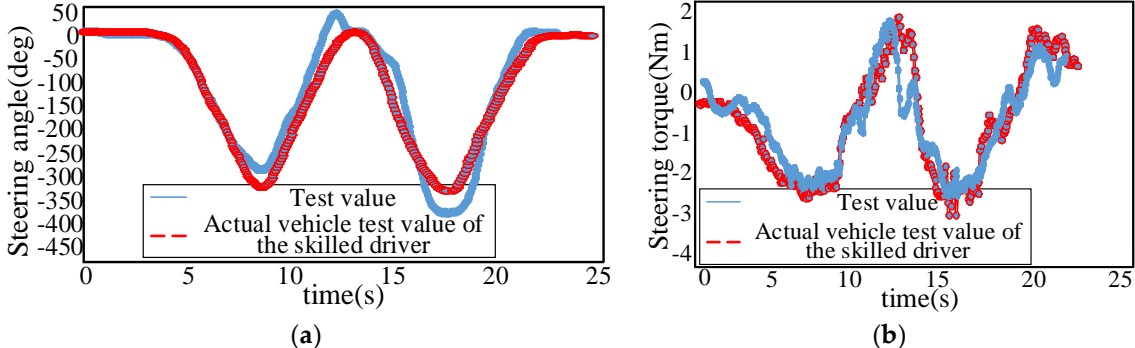

**Figure 14.** Comparison of test results under U-turn condition. (**a**) Comparison of steering wheel angle; (**b**) Comparison of steering wheel torque.

The test vehicle speed was selected as 40 km/h under the normal right-steering condition. As shown in Figure 13, the steering wheel angle obtained by the bench test could track the actual

vehicle test value of the skilled driver. The proposed intelligent vehicle human-simulated steering control system can better achieve the task of human-simulated driving.

The test vehicle speed was selected as 20 km/h under the U-turn condition. It can be seen from Figure 14 that the steering wheel angle and torque obtained by the bench test could track the actual vehicle test value of the skilled driver.

## 5. Conclusions

In this paper, the segmental polynomial was used to express the driving trajectory of the skilled driver under the two typical steering conditions, right-turn and U-turn. A novel EPS steering system with dual torque/angle sensor was proposed, and the dynamic model of the steering system was established to illustrate the intelligent controller design process. Segment control of different control modes was implemented for angle and torque. The steering system test platform based on Simulink/Carsim joint simulation was established to carry out the simulation analysis. Moreover, the intelligent vehicle steering system test bench with a steering resistance torque simulation device was constructed. The intelligent vehicle steering system platform and dSPACE rapid prototyping platform were connected to form a control system test platform and test the performance of the human-simulated steering control system. The results show that the effectiveness of the control algorithm compared with the traditional PID control has been verified from the aspects of tracking effect and comfort of passengers. The designed controller is stable and effective, and the steering angle and torque of the steering system can better track the angle and torque of the skilled drivers. Therefore, the human-simulated steering control target is realized, which lays a foundation for the development and application of the human-simulated steering control technology of intelligent vehicles.

**Author Contributions:** H.J. and H.T. conceived of and designed the method. H.T. and B.T. made software simulation and analyzation. H.T. and Y.H. performed the experiments and analyzed the experimental data. H.T. wrote the paper with the help of H.J., Y.H. and B.T.

**Funding:** This work is financially supported by The National Natural Science Fund (No. U1564201, No. 51,675,235 and No.51605199), Natural Science Fund of Jiangsu Province (No.BK20160527), and China Postdoctoral Science Fund (No.2016M590417).

**Conflicts of Interest:** The authors declare no conflict of interest.

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
