# Peer review of "Research on Control of Intelligent Vehicle Human-Simulated Steering System Based on HSIC"

_applsci, doi:10.3390/app9050905_

Round 1

Reviewer 1 Report

The paper proposes a new steering system control model based on human simulated intelligent control (HSIC). The driving of five different experienced drivers is captured under three conditions, modeled with a segmented polynomial expression and tested first in a Simulink/Carsim environment and then in a dSPACE/prototype test bench environment. The paper is clearly structured in this logical sequence and the thoroughness of the modeling is to be commended. The authors show that the new steering system model is more effective at emulating the behavior of experienced drivers than traditional methods. They also state that passenger comfort and safety is improved.

I would like to divide my feedback to the authors into a) methodology / results and b) form / style

Methodology / Results

Very good and interesting progression from mathematical model to simulated environment to prototype and very good depth of simulation of the physical environment.

It would be interesting to understand how the experienced drivers differ from each other - this information seem to be missing. Figures 1 and 2 show the average and fit the polynomial to the average. Is this the best driving behavior? Are there significant variations among drivers that should be understood and modeled?

However, it is not clear to me how the statements regarding passenger comfort have been proven (272-276 and 292-294). I understand that statements are based on figures 8c 8g 9c and 9g butI cannot detect significant differences between the yaw rates. The conclusion needs to be explained in a clearer fashion / with more effective data.

I am also having difficulty detecting the obvious control lag of PID for the U-turn condition (290-291). This is clear to me in figure 8 but not in figure 9 - at least it does not appear significant. Please explain in more detail.

It is also not clear to me how the data shown in section 3 and section 4 match with each other. I would have expected the information in figure 13 for the experienced driver to match that of figure 8, for example, but that does not seem to be the case. A better explanation of how the two are linked is needed.

Given the significance of HSIC it would be interesting to discuss in the introduction in greater detail the relevant literature, place the research more clearly in this context, highlight the contribution to this field and articulate in the conclusions what the next research steps should be to further advance the field.

Please also explain in more details the limitations of PID and how your research is addressing them. 

Please explain why you have chosen a second order polynomial fit (136) and why you are solving a system of three equations (equation 1).

Please explain why the proposed EPS is novel (172-173).

Please explain how you selected the control modes 1-5 (233-237) and how you set the relevant parameters in the simulation (254-255).

Form / Style

Figures 1 and 2 need a legend.

Line 148 - there is no yaw angle in equations 2 and 3

Line 180 the variables shown are not correct. (theta sub s and T sub m not T sub s. Also T sub w is missing)

Similarly K sub e and V sub m are missing from the explanations for equations 8 and 9 (185-188)

Figure 5 needs a legend / labeling of the axes

The terminology in the figures and text should match. For example standardize "yaw rate" in the figures, "yaw velocity" (272)  and "yaw angular velocity" (292).

To improve readability I suggest to divide the introduction into 2-3 subsections, separate modeling from results (3.1-3.3 from 3.4) and also divide 3.4 into two sections for the two conditions.

Overall, readability should be improved with further thorough editing to guarantee clear and concise English. For example: pick present or past tense in the abstract, better integrate lines 25-27. Clean up lines 31-34  (intellectualization = ?, intensity = disutilty), clean up lines 69-70, 75-76, 82-83.

line 114 - why is the driver excellent and not experienced as stated previously?

Overall, a very interesting and well-structured modeling and simulation development. An improved presentation of the method and results would facilitate understanding the significance of the research contribution. A more stringent analysis and argumentation is necessary to fully support the authors' conclusions.

Author Response

Dear reviewer

First, on behalf of other authors, I am very appreciative for giving us so many good suggestions. According to your suggestions, we decide to revise our manuscript, and submit a list of changes or rebuttals against each point raised by reviewers. The PDF file of the response to the comments has been uploaded.

Thank you very much again.

Yours sincerely,

Haobin Jiang; Huan Tian; et al.

2019-1-30

Reviewer 2 Report

A very interesting article describing the process of research on logic-based intelligent control algorithm.

The authors in a clear and understandable way introduce the reader to the automatic driving control technology of intelligent vehicle. The structure of the article text is transparent. The research methods have been clearly formulated and described and confirmed by numerical analyzes and prototype tests. The conclusions prove the effectiveness of the developed control algorithm.

The article is written on a high substantive level and contributes new elements to the established knowledge.

Author Response

Dear reviewer

On behalf of other authors, I am very appreciative for giving us so good comments.

Yours sincerely,

Haobin Jiang; Huan Tian; et al.

2019-1-30

Reviewer 3 Report

The paper presents the problem of vehicle control based on experiences of good driver skills uses nonlinear fitting of driving trajectory in chosen scenario. The problem is interesting and proper to the journal. State of the art is good. But Abstract no clear presents idea of the paper and should be corrected. The paper is written in not clear way. Sometimes it is incoherent. The results are presented without discussing how and under what conditions they were obtained, and only the conclusions from them are described. There are many repetitions in the text.

Detailed remarks:

line 14      abbreviations used for the first time require      clarification,

table      1. What means „1’’=30,7m”, usualy 1’’= 2,54 cm??

how trajectory fragments were selected for  approximation?? (e.g. fig. 1, fig. 2)

Par. 2 is not very detailed: why polynomial of 2nd order, how was  chosen fragments of trajectory????

Line 136, 137 – style,

What is      it y1, y2, y3 in (3), why 3 equations??

Eq. (2)      and (3) the lack of explanation of variables, line 148 where in eq. is „ψ”,      „v” „t01” there is no in eq. – its something wrong with explanation. The      part from line 146 up to 161 is ambiguous.

For      what is eq. (4)??? There is not used later.

Line 159      it is something wrong with time intervals,

Fig. 3      - what is the condition of choice in „case”      block??

What is „G” in (6)?

Variable are written in different way in eq. and in      text (e.g. (6) and line 180),

Line 180, where in eq. is Ts, What is it Tw??

What is it Tm, Ke, Vm in (8)

Line 195      – variable „u” earlier was „v”

The lack      of descriptions to eq. (10), (11), (12) – part 3.2 is totally unreadible,

Line 213      (and in another places) corner – angle???

Line 215      - what the error is about?? What is definition      of „e” in fig. 5,

Line 219, 221, 223 is used „deviation” of what?? Which      variable? Speed??

Line 233,      variable descritption: un, Umax, en?????

very weak description of fig. 6 and fig. 7.      What is on input fig.6?? Step of what?? The same in fig. 7,

What is      „Vehicle quality” in table 2 expressed      in kg??

Line 255, these variable and parameters have any      unites??

no description of the experiment whose results are      presented in Fig. 8

Again, no description of where the results from Fig.      9 are coming from, immediately there are some conclusions,

The part in lines 286-296, it's repeated,

There are any information, and technical data about      the system from fig. 12

Line 328-329      and 335 – how the speed is results from fig. 13 and 14???

Some positions of references need corrections (nr      of pages, e.g. pp. ….. in many positions)

the work need editorial and language corrections in      many places.

I think that the paper need major revision

Author Response

(The authors gave the same response as above.)

Round 2

Reviewer 1 Report

Dear colleagues,

thank you for the thoughtful reply to my comments and for the many changes to the paper. However, I think the main concerns I expressed have not yet been adequately addressed in the paper.

Lines 280-282 - in Figure 8(c) and 8 (g) I cannot detect significant differences for passenger comfort in yaw rate. The curves are of course different, but what is the determining factor for passenger comfort? Maximum yaw rate / rate of change of yaw rate / duration etc.? I cannot immediately tell that one behavior is more comfortable than the other. In your comments you explain the statement as a combination of factors contributing to passenger comfort, however the paper does not reflect this explanation, and there is no stated reason for this conclusion

Lines  300-302 - same comments based on Figure 9(c) and 9(g)

Lines 297-298 - I can only detect minor control lag and overshoot in the charts and am not sure that this is significant

Several other comments were addressed in the written responses, for which I am grateful, but I would like to suggest they should be addressed directly in the paper.

Author Response

Dear reviewer

First, on behalf of other authors, I am very appreciative for giving us so many good suggestions. According to your suggestions, we decide to revise our manuscript, and submit a list of changes or rebuttals against each point raised by reviewers. The PDF file of the response to the comments has been uploaded.

Thank you very much again.

Yours sincerely,

Haobin Jiang; Huan Tian; et al.

2019-2-20

Reviewer 3 Report

The paper presents the problem of vehicle control based on experiences of good driver skills uses nonlinear fitting of driving trajectory in chosen scenario. The problem is interesting and proper to the journal. State of the art is good.

But abstract is still not clear and should be corrected.

Fragments of the paper are written in not clear way. See comments and response 14 and 19. I suppose to remove this parts or written them more detailed.

-          how trajectory fragments were selected for approximation??(e.g. fig. 1, fig. 2) – this proble is stiil not clear. See fig. 1 – trajectory is in logitudinal distance in 0-130m, fragments of trajectories have about 2 m in this direction. How were thay selcected from whole trajectory??? Response 3 is not true.

-          Variables in equations and in the text are written in different ways. See eq. (3), (8), (9) and the text after,

-          I do not agree that author does not have to explain of variables as in (10), (11), (12) because they are not very important. If yes, please remove them. The same with fig. 6 and 7.

-          What is „Vehicle quality” in table 2 expressed in kg?? Whay „quality” have unit „kg”??

Author Response

(The authors gave the same response as above.)

Round 3

Reviewer 1 Report

Dear colleagues,

thank you for the revisions and the clarifications.

I think the changes pertaining to figure 8 address my concerns.

Regarding the comments pertaining to figure 9 I still cannot detect significant yaw rate differences between the two control systems and would propose a wording change to lines 284-291 as follows:

... is not obvious. Therefore, the control effect based on human-simulated intelligent control calculation is obviously better than the traditional PID control algorithm at imitating experienced drivers. The yaw rate of the artificial human intelligent control is significantly lower than the traditional yaw based angular yaw velocity value, indicating that the human-simulated intelligent control can effectively improve the comfort. It can be seen from the lateral deviation that the vehicle-based intelligent control always maintains a low lateral deviation, indicating that the  vehicle is always in a stable state. Therefore, the proposed control method can effectively improve the comfort of passengers.

Author Response

Dear reviewer

First, on behalf of other authors, I am very appreciative for giving us so many good suggestions. According to your suggestions, we decide to revise our manuscript, and submit a list of changes or rebuttals against each point raised by reviewers. The PDF file of the response to the comments has been uploaded.

Thank you very much again.

Yours sincerely,

Haobin Jiang; Huan Tian; et al.

2019-2-22

Reviewer 3 Report

In this revised version Authors took into account all the suggestions of the reviewer and explained the questions. I have no other questions.

Author Response

Dear reviewer

On behalf of other authors, I am very appreciative for giving us these comments.

Yours sincerely,

Haobin Jiang; Huan Tian; et al.

2019-2-22